# Real-world efficacy of biological agents in moderate-to-severe plaque psoriasis: An analysis of 75 patients in Taiwan

**Yu-Chen Chen**[1‡], **Yi-Ting Huang**[1‡], **Chao-Chun Yang**[1,2], **Edward Chia-Cheng Lai**[3], **Cheng-Han Liu**[1], **Chao-Kai Hsu**[1,2], **Tak-Wah Wong**[1,4,5], **Sheau-Chiou Chao**[1], **Hamm-Ming Sheu**[1], **Chaw-Ning Lee**[1,3]*

1 Department of Dermatology, National Cheng Kung University Hospital, College of Medicine, National Cheng Kung University, Tainan, Taiwan, 2 International Center for Wound Repair and Regeneration, National Cheng Kung University, Tainan, Taiwan, 3 School of Pharmacy, Institute of Clinical Pharmacy and Pharmaceutical Sciences, College of Medicine, National Cheng Kung University, Tainan, Taiwan, 4 Department of Biochemistry and Molecular Biology, College of Medicine, National Cheng Kung University, Tainan, Taiwan, 5 Center of Applied Nanomedicine, National Cheng Kung University, Tainan, Taiwan

‡ These authors co-first authors on this work.
* joyce060324@gmail.com

**Data Availability Statement:** All relevant data are within the manuscript.

**Funding:** The work was funded by a grant (MOST 109-2628-B-006 -035) from the Ministry of

## Abstract

### Background

Real-world clinical data on psoriasis patients receiving different biological agents is needed, especially in Asian populations.

### Objectives

Our aim is to compare and analyze the efficacy and safety profile of four biological agents (etanercept, adalimumab, ustekinumab and secukinumab) in a real-world setting in Taiwan.

### Methods

We retrospectively analyzed the clinical data of all patients with moderate-to-severe plaque psoriasis (Psoriasis Area and Severity Index (PASI) ≥ 10) who received etanercept, adalimumab, ustekinumab or secukinumab between January 2011 and December 2018 in a tertiary hospital in Taiwan.

### Results

A total of 119 treatment episodes in 75 patients were included in this study. Ustekinumab was used in 49 treatment episodes, followed by secukinumab in 46 treatment episodes, adalimumab in 14 treatment episodes and etanercept in 10 treatment episodes. The proportion of the biologic-naïve was highest in etanercept (100%) and lowest in secukinumab (23.9%). The PASI-75, -90 and -100 were the highest in secukinumab (91.3%, 82.6%, 41.3%, respectively), followed by ustekinumab (79.6%, 44.9%, 16.3%), adalimumab (64.3%, 28.6%, 7.1%) and etanercept (50.0%, 30.0%, 0%). The rate of adverse events that required treatment was highest for secukinumab (15.2%), followed by adalimumab (14.3%),

Science and Technology, Taiwan to CCY. The
funder had no role in study design, data collection
and analysis, decision to publish, or preparation of
the manuscript.

**Competing interests:** The authors have declared
that no competing interests exist.

ustekinumab (8.2%), and etanercept (0%), including 4 cases of infections, 2 cases of cardio-
vascular diseases and 4 cases of cancers.

## Conclusions

This real world data showed differential efficacy and safety of the four biological agents.

## Introduction

Psoriasis is a common, chronic and inflammatory skin disease, affecting approximately 2–3%
of the world population [1]. Plaque psoriasis, or psoriasis vulgaris, is the most frequent clinical
variant (approximately 85 to 90%) of this disease [2], which manifest as raised, well-demar-
cated, erythematous and oval plaques with adherent silvery white scales [1]. Moderate-to-
severe psoriasis usually requires systemic treatment such as phototherapy, retinoids, metho-
trexate, cyclosporine, or biological agents [3]. Over the past decades, the introduction of bio-
logical agents has remarkably changed the treatment of moderate-to-severe psoriasis and
psoriatic arthritis [4]. Today, for patients with inadequate response, contraindication or intol-
erable adverse effects to traditional systemic medication, biological agents represent an impor-
tant and effective therapeutic tool [4]. Along with the progress in understanding of the
molecular and immunologic basis of psoriasis, highly targeted biological agents have since
developed with the perspective not only to improve but to clear psoriasis [5]. In recent years,
many randomized clinical trials and real-life study have already shown the good efficacy and
safety of biological agents [6–13]. However, more real-world data are needed to reflect the clin-
ical situations without undergoing strict patient selection and monitoring [14]. Besides, real-
world therapeutic data of biological agents of non-Caucasian populations is currently limited
[15]. In this study, the objective is to compare the efficacy and safety profiles of four different
biological agents, etanercept (TNF-α inhibitor), adalimumab (TNF-α inhibitor), ustekinumab
(IL-12/IL-23 inhibitor) and secukinumab (IL-17 inhibitor), in a real-world setting in Taiwan.

## Methods

### Patients and data collection

The study protocol was approved by the Institutional Review Board of National Cheng Kung
University Hospital (No. A-ER-108-163). No consent was obtained, because the data were ana-
lyzed anonymously. The researchers anonymized the data after obtaining the data. All data
was collected from the electronic medical record system and the electronic imaging database
of the Department of Dermatology.

For the cohort selection, we included all patients with moderate-to-severe psoriasis vulgaris
(defined as Psoriasis Area and Severity Index (PASI) [16] $\geq$ 10) who received etanercept, adali-
mumab, ustekinumab or secukinumab between January 2011 and December 2018 at the
Department of Dermatology, National Cheng Kung University Hospital, Tainan, Taiwan, and
the databases was accessed on July 25, 2019. All the biological treatments were approved and
fully reimbursed by National Health Insurance in Taiwan. The prescription of all biological
agents complied with the recommended dose and schedule. For etanercept, the starting dose
was 50 mg twice weekly for 3 months and the maintenance dose was 25 mg twice weekly. For
adalimumab, the initial dose was 80 mg, followed by 40 mg since week 1. For ustekinumab, 45
mg was administered initially at week 0, followed by 45 mg every 12 weeks since week 4, in
patients with body weight less than 100 kg. The dose of ustekinumab for patients with body

weight greater than 100 kg was doubled. For secukinumab, 300 mg SC was the initial dose at weeks 0, 1, 2, 3, and 4, followed by monthly 300mg beginning at week 8. Other systemic treatments for psoriasis and phototherapy had to be discontinued within two months after the initiation of the biological agents according to the regulation of National Health Insurance in Taiwan. Patients with incomplete serial PASI score data, ie. more than one PASI score missed, were excluded. There were some cases who paid the expenses of biological treatment by themselves, but these cases were also excluded due to the irregular dosing regimen. One patient could have more than one treatment episodes if he or she received more than one biological agent.

## Data analysis

The demographic and clinical data (age, sex, body mass index (BMI), presence/absence of arthralgia, age of diagnosis, duration of disease, comorbidities and previous biological agents use) of the patients were collected and analyzed. The severity of psoriasis vulgaris was evaluated using PASI score before the biological treatment (as baseline PASI score) and at each clinic visit according to the different administration protocols of the 4 biological agents. For the treatment efficacy, it was analyzed by the best improvement (%) of PASI score, PASI-75, PASI-90, and PASI-100. Drug survival was defined as the total treatment duration of a certain biological agent until discontinuation or switch to another agent. The drug survival, time to achieve best PASI score and percentage of discontinuation were also analyzed and compared among treatment groups. Adverse events (AEs) that required medical treatment or follow-up during biological agent use were recorded at each visit. The case number (n) throughout the article indicates the numbers of treatment episodes, rather than numbers of patients, unless specified otherwise.

## Statistical analysis

All analysis was performed with Prism 8 and SAS, Version 9.3. Data were presented as mean for continuous variables and number and percentage for categorical variables. One-way analysis of variance (ANOVA) and Chi-squared test were applied to compare baseline differences in the four treatment groups for continuous variables. The effects between biologics was compared by multivariate generalized estimating equations (GEE) models with repeated measures of patients for each outcome indicators, including the best PASI score, PASI-75, -90 and -100. PASI was treated as a continuous variable while analyzing the best PASI score. The outcomes of PASI-75, -90 or -100 were regarded as binary categories with PASI-75, -90 or -100 as the cut-off value, respectively. The age, sex, baseline PASI, BMI and bio-naïve/experienced were included in the models. Student's t-test was used to compare between the bio-naïve subgroup and the bio-experienced subgroup.

## Results

### Patient characteristics

A total of 119 treatment episodes, which were administered in 75 patients, were included in this study. Detailed demographic data was presented in Table 1. Among all the study population, the mean age was 44.0 years. The number of treatment episodes in male patients outnumbered female patients (101 vs.18 treatment episodes). The mean age of diagnosis of psoriasis was 28.3 years and mean disease duration was 15.8 years. The mean BMI was 27.6 kg/m$^2$, which fell into the category of obesity for Taiwanese population. The mean PASI score at baseline was 23.1, which indicated that the severity of psoriasis of the patients included in this

**Table 1. Demographic data and disease characteristics of the study population.**

| | Etanercept (n = 10) | Adalimumab (n = 14) | Ustekinumab (n = 49) | Secukinumab (n = 46) | Total (n = 119) |
|---|---|---|---|---|---|
| **Sex** | | | | | |
| **Men, n (%)** | 9 (90.0) | 12 (85.7) | 41 (83.7) | 39 (84.8) | 101 (84.9) |
| **Women, n (%)** | 1 (10.0) | 2 (14.3) | 8 (16.3) | 7 (15.2) | 18 (15.1) |
| **Age, mean (range), years** | 38.7 (13–49) | 40.5 (22–66) | 43.3 (14–67) | 46.8 (25–41) | 44.0 (13–67) |
| **Age of diagnosis, mean, years** | 23.9 | 26.9 | 28.0 | 29.8 | 28.3 |
| **BMI at baseline, mean, kg/m$^2$** | 28.6 | 27.7 | 27.6 | 27.3 | 27.6 |
| **Patient with arthralgia, n (%)** | 7 (70.0) | 10 (71.4) | 23 (46.9) | 24 (52.2) | 63 (53.3) |
| **Disease duration, mean, years** | 14.8 | 13.6 | 15.3 | 17.0 | 15.8 |
| **Baseline PASI, mean (range)** | 24.0 (15.2–32.8) | 24.2 (10.0–39.6) | 25.4 (10.4–126.6) | 20.2 (10.5–43.1) | 23.1 (10.0–126.6) |

BMI, body mass index; PASI, Psoriasis Area Severity Index.

The comparison between the 4 groups was performed by ANOVA or Chi-square test.

study was high. There were no significant differences in terms of age, disease duration, age of diagnosis, BMI, and baseline PASI score between the four treatment groups (Table 1).

Among all the treatment episodes, there was 52.5% biologic-naïve. In the etanercept group, all the treatment episodes were biologic-naïve since it was the first biological agent approved for psoriasis treatment by National Health Insurance in Taiwan. The most recently approved biological agent, secukinumab, had the lowest percentage of biologic-naïve series (23.9%) (Table 2).

## Efficacy of etanercept, adalimumab, ustekinumab and secukinumab

The treatment efficacy of the four biological agents was summarized in Table 3. In general, the data showed that secukinumab had the lowest "best PASI score", shortest "time to achieve best PASI", best rate of PASI-75, PASI-90 and PASI-100 (Table 3). On the other hand, etanercept had highest "best PASI score", lowest rate of PASI-75, -90 and -100, while adalimumab had the longest "time to achieve best PASI".

Using GEE models, the chance of achieving PASI-75 was higher in secukinumab patients, compared to ustekinumab (OR = 4.9, P = 0.004), adalimumab (OR = 13.8, P = 0.0002) and etanercept (OR = 28.6, P = 0.0001) (Table 4). For PASI-90, secukinumab patients were associated with greater chance of achieving it, compared to ustekinumab (OR = 20.2, P<0.0001), adalimumab (OR = 89.2, P<0.0001) and etanercept (OR, 58.6, P = 0.0007). For PASI-100, secukinumab patients were associated with greater chance of achieving it, compared to ustekinumab (OR = 5.5, P = 0.003). The result of GEE models was listed in detail in Table 4.

We further analyzed and compared the efficacy on treatment episodes with and without exposure to previous biological agents (bio-naïve and bio-experienced) (Table 5). In general, the biologic-naïve group had better treatment response than the biologic-experienced group in terms of best PASI score, PASI-75, PASI-90 and PASI 100 in all biological agents.

**Table 2. The numbers of treatment series receiving first-line therapy to forth-line therapy in the four treatment groups.**

| | Etanercept (n = 10) | Adalimumab (n = 14) | Ustekinumab (n = 49) | Secukinumab (n = 46) | Total (n = 119) |
|---|---|---|---|---|---|
| **First-line, n (%)** | 10 (100.0) | 10 (71.4) | 31 (63.3) | 11 (23.9) | 62 (52.5) |
| **Second-line, n (%)** | 0 (0.0) | 3 (21.4) | 16 (32.7) | 20 (43.5) | 38 (32.2) |
| **Third-line, n (%)** | 0 (0.0) | 1 (7.1) | 2 (4.1) | 11 (23.9) | 14 (11.9) |
| **Forth-line, n (%)** | 0 (0.0) | 0 (0.0) | 0 (0.0) | 4 (8.7) | 4 (3.4) |

**Table 3. The summary of mean PASI score changes, time to achieve best PASI and proportion of PASI-75, PASI-90 and PASI-100.**

|  | Etanercept (n = 10) | Adalimumab (n = 14) | Ustekinumab (n = 49) | Secukinumab (n = 46) | Total (n = 119) |
|---|---|---|---|---|---|
| **Baseline PASI, mean** | 24.0 | 24.2 | 25.4 | 20.2 | 23.1 |
| **Best PASI ever achieved, mean** | 10.6 | 6.9 | 3.9 | 1.6 | 3.9 |
| **Time to achieve best PASI, mean, weeks** | 48.6 | 70.4 | 39.1 | 20.0 | 36.2 |
| **PASI-75, n (%)** | 5 (50.0) | 9 (64.3) | 39 (79.6) | 42 (91.3) | 95 (79.8) |
| **PASI-90, n (%)** | 3 (30.0) | 4 (28.6) | 22 (44.9) | 38 (82.6) | 67 (56.3) |
| **PASI-100, n (%)** | 0 (0.0) | 1 (7.1) | 8 (16.3) | 19 (41.3) | 28 (23.5) |

PASI, Psoriasis Area Severity Index.

## Adverse events of etanercept, adalimumab, ustekinumab and secukinumab

The AEs that required medical treatment or follow-up during biological agents use were analyzed (Table 6). AEs were further categorized into four groups: infection, cardiovascular diseases, neoplasm and others. There was no severe infection that led to long-term discontinuation of biological therapy or death. There were two patients with herpes zoster infection; one in ustekinumab group and one in secukinumab group. Besides, there was one patient having hepatitis B reactivation during ustekinumab treatment. One patient had folliculitis on scalp with lymphadenopathy during secukinumab treatment. There was no case of candida infection or tuberculosis reactivation. There were two major cardiovascular adverse events during treatment of secukinumab. There were three malignancies which led to long-term suspension of biological agents; one patient diagnosed with esophageal cancer during ustekinumab treatment, one patient diagnosed with rectal cancer and one diagnosed with hepatocellular carcinoma during secukinumab treatment.

## Discussions

This single-center retrospective study presents real-world data on drug efficacy, and AEs in patients with moderate-to-severe psoriasis using etanercept, adalimumab, ustekinumab and secukinumab in Taiwan. To date, there are limitedreal-world studies comparingdrug efficacy and safety profile of etanercept, adalimumab, ustekinumab and secukinumab together [17, 18]. Our result is comparable with the head-to-head trials and real-life studies [6–13]. Secukinumab had the highest rate of PASI-75, PASI-90 and PASI-100, followed by ustekinumab, adalimumab and etanercept. It is remarkable that secukinumab showed the best efficacy in reaching PASI-75, PASI-90 and PASI-100, despite that this group had highest proportion of the biologic-experienced (76.0%). However, because of shorter follow-up for patients receiving secukinumab compared to other three drugs, and most of the treatments are still ongoing

**Table 4. The comparison of the rate of PASI-75, PASI-90 and PASI-100 in four biological agents by GEE models.**

|  | SEC vs. UST | SEC vs. ADA | SEC vs. ETA | UST vs. ADA | UST vs. ETA | ADA vs. ETA |
|---|---|---|---|---|---|---|
| **PASI-75, OR** | 4.9 (P = 0.004) | 13.8 (p = 0.0002) | 28.6 (p = 0.0001) | 2.8 (p = 0.16) | 5.8 (p = 0.06) | 2.1 (p = 0.43) |
| **PASI-90, OR** | 20.2 (p<0.0001) | 89.2 (p<0.0001) | 58.6 (p = 0.0007) | 4.4 (p = 0.04) | 2.9 (p = 0.31) | 0.6 (p = 0.7) |
| **PASI-100, OR** | 5.5 (p = 0.003) | N/A[†] | N/A[†] | N/A[†] | N/A[†] | N/A[†] |

ADA, adalimumab; ETA, etanercept; SEC, secukinumab; UST, ustekinumab.

N/A, not available; OR, odds ratio; PASI, Psoriasis Area Severity Index.

[†]Data not available due to the small case numbers in PASI-100 patients of the adalimumab and etanercept groups which could not be applied in GEE models.

**Table 5. The comparison of efficacy between the bio-naïve and the bio-experienced.**

|  |  | Etanercept (n = 10) | Adalimumab (n = 14) | Ustekinumab (n = 49) | Secukinumab (n = 46) |
|---|---|---|---|---|---|
| **Case number, n (%)** | Bio-naïve | 10 (100.0) | 10 (71.4) | 31 (63.3) | 11 (23.9) |
|  | Bio-experienced | 0 (0) | 4 (28.6) | 18 (36.7) | 35 (76.1) |
| **Baseline PASI, mean** | Bio-naïve | 24.0 | 23.9 | 25.9 | 19.9 |
|  | Bio-experienced | N/A | 24.9 | 24.6 | 20.2 |
| **Best PASI ever achieved, mean** | Bio-naïve | 10.6 | 5.1 | 3.1 | 1.2 |
|  | Bio-experienced | N/A | 11.1 | 5.5 | 1.7 |
| **PASI-75, n (%)** | Bio-naïve | 5 (50.0) | 8 (80.0) | 27 (81.8) | 11 (100.0) |
|  | Bio-experienced | N/A | 1 (25.0) | 12 (66.7) | 31 (88.6) |
| **PASI-90, n (%)** | Bio-naïve | 3 (30.0) | 4 (40.0) | 18 (58.1*) | 11 (100.0) |
|  | Bio-experienced | N/A | 0 (0.0) | 4 (22.2*) | 27 (77.1) |
| **PASI-100, n (%)** | Bio-naïve | 0 (0.0) | 1 (10.0) | 7 (22.6) | 6 (54.5) |
|  | Bio-experienced | N/A | 0 (0.0) | 1 (5.6) | 13 (37.1) |

N/A, not available, PASI, Psoriasis Area Severity Index.

*p<0.05, bio-naïve vs. bio-experienced, t-test.

(52.2%), we should interpret our result with great caution. Longer follow-up for recurrence or adverse events should be done in the future.

In line with other studies [19, 20], biologic-naïve subgroup had better treatment response in our study. However, the reasons why previous biological agents use is a predictor of worse efficacy remains uncertain and further investigation is warranted [19].

The drug survival time, a commonly used indicator for the efficacy of the biological agents, may not be a good indicator for Taiwanese population, because the treatment period was greatly influenced by the insurance policy which terminates the reimbursement of biological agents after 2 years. The National Health Insurance system in Taiwan covers the full expenses of biological treatment for moderate to severe psoriasis patients (PASI ≥10) and the prerequisite criteria are lack of efficacy of two conventional medications and phototherapy [21, 22]. Biological agents are provided up to 2 years as long as PASI-50 is reached. After a 2-year treatment, if the PASI score is below 10, biological agents will not be reimbursed until there is a relapse with PASI ≥10. Drug survival was also influenced by the timing when the individual biological agent was available. Therefore, we should interpret our data with caution and also consider the differences in the frequency of dosing, market entry times, duration of follow-up and reimbursement regulations.

There was no severe infection leading to long-term drug discontinuation. There were three malignancies occurring during ustekinumab or secukinumab treatment and all resulted in

**Table 6. Adverse events (AEs) during the use of biological agents.**

|  | Etanercept (n = 10) | Adalimumab (n = 14) | Ustekinumab (n = 49) | Secukinumab (n = 46) |
|---|---|---|---|---|
| **All AE that requires treatment or follow-up, n (%)** | 0 (0.0) | 2 (14.3) | 4 (8.2) | 7 (15.2) |
| • Infection, n (%) | 0 (0.0) | 1 (7.1) | 2 (4.1) | 3 (6.5) |
| • Cardiovascular diseases, n (%) | 0 (0.0) | 0 (0.0) | 0 (0.0) | 2 (4.3) |
| • Neoplasm, n (%) | 0 (0.0) | 1 (7.1) | 1 (2.0) | 2 (4.3) |
| • Others†, n (%) | 0 (0.0) | 0 (0.0) | 1 (0.0) | 0 (0.0) |

AE, adverse event.

†The AE was liver function impairment.

long-term withdrawal of biologicaltreatment. This result showed that safety was still a concern in around 10% of the patients under the treatment of biological agents in a real-life setting.

The strengths of our study included complete recording of serial PASI scores of each patient and standard dosage and fixed injection schedule of biological treatment in order to meet the criteria of reimbursement of National Health Insurance. It provided real-world therapeutic data of non-Caucasian ethnic populations. The main limitations of this study were its retrospective nature and non-comparative study design. The patients were not randomized or blinded during treatment course. Besides, the sample size of this study is small, with 119 treatment episodes. The follow-up period was too short for the latest available biological agent, secukinumab.

In conclusion, our study provides real-life clinical experience on biological treatment for psoriasis in Taiwan. The results showed differential efficacy of the four biological agents. Special attention still needs to be paid on AEs which occur during the use of biological agents, including infection, cardiovascular diseases and cancers.

## Acknowledgments

We would like to thank the participants for their willingness to contribute data that make analyses such as these possible, and also appreciate the staff and administrators for their work.

## Author Contributions

**Formal analysis:** Chao-Chun Yang, Edward Chia-Cheng Lai.

**Investigation:** Yu-Chen Chen, Chao-Chun Yang, Cheng-Han Liu, Chao-Kai Hsu, Tak-Wah Wong, Sheau-Chiou Chao, Hamm-Ming Sheu, Chaw-Ning Lee.

**Methodology:** Chao-Chun Yang, Chaw-Ning Lee.

**Supervision:** Chao-Chun Yang.

**Writing – original draft:** Yi-Ting Huang, Chaw-Ning Lee.

**Writing – review & editing:** Yu-Chen Chen, Yi-Ting Huang, Chao-Chun Yang, Chaw-Ning Lee.

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
