## [Decision Letter · Decision Letter 0]

12 Oct 2020

PONE-D-20-24855

Real-world efficacy of biological agents in moderate-to-severe plaque psoriasis: an analysis of 75 patients in Taiwan

PLOS ONE

Dear Dr. Lee,

Thank you for submitting your manuscript to PLOS ONE. After careful consideration, we feel that it has merit but does not fully meet PLOS ONE’s publication criteria as it currently stands. Therefore, we invite you to submit a revised version of the manuscript that addresses the points raised during the review process.

I am returning your manuscript with three reviews. The reviewers came to different conclusions about the paper, as you will see. After reading the reviews and looking at the manuscript, I have to concur with the more critical review. I am sorry I cannot be more positive at the moment, but as I have noted, all is not lost. It requires a lot of work and a major revision that I believe that you need more time to work on the manuscript for a resubmission if you wish to do so.

We look forward to receiving your revised manuscript.

Kind regards,

Vineet Kumar Rai, PhD

Academic Editor

PLOS ONE

Journal Requirements:

2. Thank you for stating in your ethics statement in the online submission form "No consent was obtained, because the data were analyzed anonymously." Please clarify whether all data were fully anonymized before you accessed them or if the researchers anonymized the data after they obtained the data.

In addition, it appears from your ethics document that the ethics committee waived the requirement for informed consent. If so, please add this statement to your ethics statement.

3. Thank you for stating in the text of your manuscript "The study protocol was approved by the Institutional Review Board of National Cheng Kung University Hospital." Please also add this information to your ethics statement in the online submission form.

4. Please provide a reference for the PASI score.

5. Please include the date(s) on which you accessed the databases or records to obtain the data used in your study.

6. We note you have included a table to which you do not refer in the text of your manuscript. Please ensure that you refer to Table 2 in your text; if accepted, production will need this reference to link the reader to the Table.

Reviewers' comments:

Reviewer's Responses to Questions

**Comments to the Author**

1. Is the manuscript technically sound, and do the data support the conclusions?

Reviewer #1: No

Reviewer #2: Partly

Reviewer #3: Yes

2. Has the statistical analysis been performed appropriately and rigorously? 

Reviewer #1: I Don't Know

Reviewer #2: Yes

Reviewer #3: No

3. Have the authors made all data underlying the findings in their manuscript fully available?

Reviewer #1: Yes

Reviewer #2: Yes

Reviewer #3: No

4. Is the manuscript presented in an intelligible fashion and written in standard English?

Reviewer #1: Yes

Reviewer #2: Yes

Reviewer #3: Yes

5. Review Comments to the Author

Reviewer #1: This is a very small sample size with a short follow-up time, so it's difficult to make any conclusions. I think this paper would be a better fit for a psoriasis journal.

Reviewer #2: The Manuscript “Real-world efficacy of biological agents in moderate-to-severe plaque psoriasis: an analysis of 75 patients in Taiwan” is written in presentable manner. However, the number of patients is very less. Manuscript can be accepted after minor revision, if authors agreed to satisfy following comments.

Q1. Please clarify annotations “*” and “†” used in table 4 and 3 to avoid confusion. Same sign is also used for author’s equal contribution.

Reviewer #3: 1. Though the article is well written and presented, the sample size could have been large enough.

2. Line no. 88 and 89, authors have written that “the prescription of all biological agents complied with the recommended dose and schedule”.

3. The dose and schedule of the dosing should have been specified in the manuscript along with the details of the injections and make.

4. Data to support the adverse events cases is inadequate with a short follow-up time.

5. However, the discussion part needs to be extensively re-written because there are some major problems with the information presented.

6. It would be more transparent if the authors provide information on how the authors search and correlate the discussion part.

7. I understand that this manuscript is based on clinical studies, however, authors should cite more scientific evidence with proper updated citation.

8. Since the authors mentioned that the effect of the selected drugs for the treatment of mild-to-moderate psoriasis patients the information there should therefore specific about psoriasis as a combined regimen.

9. The first half of the conclusion seems irrelevant at some level and main content.

10. The author must apply some more statistical tools for the confirmation of evidence in the experiment part.

6. PLOS authors have the option to publish the peer review history of their article (what does this mean?). If published, this will include your full peer review and any attached files.

Reviewer #1: No

Reviewer #2: No

Reviewer #3: No

---

## [Author Response · Author response to Decision Letter 0]

20 Nov 2020

Answers to Review Comments to the Author

Reviewer 1#

This is a very small sample size with a short follow-up time, so it's difficult to make any conclusions. I think this paper would be a better fit for a psoriasis journal.

Reply from authors: We agree that the sample size of this study was relatively small. However, there were several strengths of this work, as stated in the Discussion section. The major strength which made us advantageous to other real world study was that the efficacy of the four biological agents under the standard regimens could be compared in a real-work setting due the reimbursement policy of the National Health Insurance in Taiwan.

Reviewer 2#

The Manuscript “Real-world efficacy of biological agents in moderate-to-severe plaque psoriasis: an analysis of 75 patients in Taiwan” is written in presentable manner. However, the number of patients is very less. Manuscript can be accepted after minor revision, if authors agreed to satisfy following comments.

Reply from authors: Thank you for the comments.

Q1. Please clarify annotations “*” and “†” used in table 4 and 3 to avoid confusion. Same sign is also used for author’s equal contribution.

Reply from authors: Thank you for the comments. The annotations were adjusted accordingly.

Reviewer 3#

1. Though the article is well written and presented, the sample size could have been large enough.

Reply from authors: We agree that the sample size of this study was relatively small. However, there were several strengths of this work, as stated in the Discussion section. The major strength which made us advantageous to other real world study was that the efficacy of the four biological agents under the standard regimens could be compared in a real-work setting due the reimbursement policy of the National Health Insurance in Taiwan. 

2. Line no. 88 and 89, authors have written that “the prescription of all biological agents complied with the recommended dose and schedule”.

Reply from authors: Thank you for the suggestion. The detail of the dose and schedule is added at line 94-103.

3. The dose and schedule of the dosing should have been specified in the manuscript along with the details of the injections and make.

Reply from authors: Thank you for the suggestion. The detail of the dose and schedule is added at line 94-103.

4. Data to support the adverse events cases is inadequate with a short follow-up time.

Reply from authors: Thank you for the comments. We agree that the follow-up period was relatively short. As stated in the Discussion, a study with longer follow-up period will be conducted to better reflect the occurrence of AE.

5. However, the discussion part needs to be extensively re-written because there are some major problems with the information presented.

Reply from authors: Thank you for the comments. The manuscript has been revised extensively in the text, tables and references.

6. It would be more transparent if the authors provide information on how the authors search and correlate the discussion part.

Reply from authors: The references were searched on PubMed by the name of the four biologicals and “real-world study” or “clinical trials”. The discussion part has been revised and updated references are added. 

7. I understand that this manuscript is based on clinical studies, however, authors should cite more scientific evidence with proper updated citation.

Reply from authors: Thank you for the suggestion. The discussion part has been revised and updated references are added. 

8. Since the authors mentioned that the effect of the selected drugs for the treatment of mild-to-moderate psoriasis patients the information there should therefore specific about psoriasis as a combined regimen.

Reply from authors: Traditional systemic treatments have to be discontinued within two months after initiation of the biological agents according to the regulation of National Health Insurance in Taiwan. Therefore the effect contributed by concomitant use of traditional biological agents could be neglected. A relevant statement is added into the Method section (Line 101-103). 

9. The first half of the conclusion seems irrelevant at some level and main content.

Reply from authors: Thank you for the comments. The Conclusion has been revised to better fit the main content of this study. 

10. The author must apply some more statistical tools for the confirmation of evidence in the experiment part.

Reply from authors: Thank you for the suggestion. A new statistic tool, multivariate generalized estimating equations (GEE) models was used for analysis. Relevant description has been added into the Methods and Results sections (Line 129-135, Line 175-182 and Table 4).

---

## [Decision Letter · Decision Letter 1]

14 Dec 2020

Real-world efficacy of biological agents in moderate-to-severe plaque psoriasis: an analysis of 75 patients in Taiwan

PONE-D-20-24855R1

Dear Dr. %Lee%,

We’re pleased to inform you that your manuscript has been judged scientifically suitable for publication and will be formally accepted for publication once it meets all outstanding technical requirements.

Kind regards,

Vineet Kumar Rai, PhD

Academic Editor

PLOS ONE

Additional Editor Comments (optional):

Reviewers' comments:

Reviewer's Responses to Questions

**Comments to the Author**

1. If the authors have adequately addressed your comments raised in a previous round of review and you feel that this manuscript is now acceptable for publication, you may indicate that here to bypass the “Comments to the Author” section, enter your conflict of interest statement in the “Confidential to Editor” section, and submit your "Accept" recommendation.

Reviewer #2: All comments have been addressed

Reviewer #3: All comments have been addressed

2. Is the manuscript technically sound, and do the data support the conclusions?

Reviewer #2: Yes

Reviewer #3: Yes

3. Has the statistical analysis been performed appropriately and rigorously? 

Reviewer #2: Yes

Reviewer #3: Yes

4. Have the authors made all data underlying the findings in their manuscript fully available?

Reviewer #2: Yes

Reviewer #3: Yes

5. Is the manuscript presented in an intelligible fashion and written in standard English?

Reviewer #2: Yes

Reviewer #3: Yes

6. Review Comments to the Author

Reviewer #2: (No Response)

Reviewer #3: I think the authors have made appropriate responses to my previous comments and concern. Now It is in a good shape for publication.

7. PLOS authors have the option to publish the peer review history of their article (what does this mean?). If published, this will include your full peer review and any attached files.

Reviewer #2: No

Reviewer #3: **Yes: **Dr. Alok Sharma

---

## [Editor Report · Acceptance letter]

16 Dec 2020

PONE-D-20-24855R1 

Real-world efficacy of biological agents in moderate-to-severe plaque psoriasis: an analysis of 75 patients in Taiwan 

Dear Dr. Lee:

I'm pleased to inform you that your manuscript has been deemed suitable for publication in PLOS ONE. Congratulations! Your manuscript is now with our production department. 

Kind regards, 

on behalf of

Dr. Vineet Kumar Rai 

Academic Editor

PLOS ONE